# Effects of Intraoperative Nefopam on Catheter-Related Bladder Discomfort in Patients Undergoing Robotic Nephrectomy: A Randomized Double-Blind Study

**DOI:** 10.3390/jcm8040519

**Published:** 2019-04-16

**Authors:** Chi-Bum In, Young-Tae Jeon, Ah-Young Oh, Se-Jong Jin, Byeong-Seon Park, Eun-Su Choi

**Affiliations:** 1Department of Anesthesiology and Pain Medicine, Konyang University Hospital, Daejeon 35365, Korea; cb523@naver.com; 2Department of Anesthesiology and Pain Medicine, Seoul National University Bundang Hospital, Seongnam 13620, Korea; ytjeon@snubh.org (Y.-T.J.); oay1@snubh.org (A.-Y.O.); 3Department of Anesthesiology and Pain Medicine, Seoul National University College of Medicine, Seoul 03080, Korea; 4Department of Anesthesiology and Pain Medicine, Korea University Ansan Hospital, Gyeonggi-do 15355, Korea; holicer_90@naver.com (S.-J.J.); pbskumc57@gmail.com (B.-S.P.)

**Keywords:** catheter-related bladder discomfort (CRBD), nefopam, postoperative pain management

## Abstract

Catheter-related bladder discomfort (CRBD) is one of the most difficult symptoms during the postoperative period. Nefopam is a non-narcotic analgesic agent, which also has anticholinergic action. This study was performed to evaluate the effects of nefopam on CRBD in male patients undergoing robotic nephrectomy. A total of 109 male patients were randomly divided into two groups: the control group (*n* = 55) received 20 mL of normal saline, and the nefopam group (*n* = 54) received 20 mg of nefopam 1 h before the end of the operation. At postoperative times of 20 min, 1 h, 2 h, and 6 h, the severity of CRBD was measured using an 11-point numeric rating scale, respectively. The severity of CRBD in the nefopam group was significantly lower than that in the control group at 20 min (4.8 ± 1.3 vs. 2.3 ± 1.0, respectively, *p* = 0.012) and at 1, 2, and 6 h (3.5 ± 1.2, 2.7 ± 0.9, and 2.5 ± 1.0 vs. 4.1 ± 0.8, 1.6 ± 0.8, and 1.3 ± 0.6, respectively, *p* < 0001). Intraoperative nefopam administration reduced the severity of CRBD in patients undergoing robotic nephrectomy.

## 1. Introduction

The Catheter-related bladder discomfort (CRBD) induced by urethral catheterization is defined as an unpleasant and burning sensation in the urethra and suprapubic area, causing an impulse to void and urinary frequency [1,2,3]. When the patient is ambulating for recovery after surgery, the catheter can severely irritate the bladder, causing pain and discomfort, which may delay the patient’s recovery. The incidence of CRBD is 58–80% [2,4,5] and moderate to severe CRBD occurs predominantly in males [6]. CRBD is caused by involuntary contraction of the bladder, which involves the M3 receptors in the bladder wall near the catheter [7].

Previous studies have shown that anticholinergic agents, such as tolterodine, oxybutynin, tramadol, and butylscopolamine, have a preventive effect on CRBD [1,5,7,8]. The non-sedative benzoxazocine analgesic nefopam has anticholinergic and sympathomimetic activity [9,10,11]. We hypothesized that the anticholinergic effect of nefopam may reduce the severity of CRBD. This randomized, controlled trial was performed to evaluate the effects of intraoperative nefopam on CRBD in patients undergoing robotic nephrectomy.

## 2. Materials and Methods

This study was approved by the Seoul National University Bundang Hospital Institutional Review Board and was registered at ClinicalTrials.gov (NCT03130010). All patients provided written informed consent forms before enrollment. The study population consisted of only male patients aged 20–75 years old with American Society of Anesthesiologists physical status I or II, who were scheduled to undergo unilateral robotic nephrectomy under general anesthesia requiring a new Foley catheter (>16 or 18 Fr) during surgery. The exclusion criteria were as follows: irritable or neurogenic bladder; previous history of a central nervous system or neurological disorder (e.g., epilepsy or patients receiving monoamine oxidase inhibitors); and serious cardiovascular disease, urethral and prostate disorders, or closed angle glaucoma. A total of 110 eligible patients were randomly assigned to the nefopam group (*n* = 55) or control group (*n* = 55) using a computer-generated block randomization list (http://www.randomization.com). An anesthesia nurse, who was not involved in the clinical care of the patients, prepared and administered the study drug according to the randomization list. To ensure blinding, the study drug and control had the same label.

All patients received intravenous midazolam at 0.03 mg/kg as a premedication to relieve preoperative anxiety. On arrival at the operating room, monitoring was started with pulse oximetry, non-invasive blood pressure monitoring, and electrocardiography. The bispectral index (BIS) (A-2000 BISTM monitor; Aspect Medical Systems, Inc., Natick, MA) was also monitored. Anesthesia was induced by the administration of propofol (2 mg/kg), continuous infusion of remifentanil (target effect-site concentration controlled infusion 3 ng/mL; Orchestra infusion pumps; Fresenius Vial, Brezins, France) and rocuronium (0.6 mg/kg), and maintained with desflurane (6–8 vol%) and continuous infusion of remifentanil (target effect-site concentration controlled infusion 2–3 ng/mL). The desflurane and remifentanil concentrations were adjusted to maintain the BIS at 40–60. During surgery, rocuronium (0.15 mg/kg) was administered as needed. The lungs were mechanically ventilated with oxygen and medical air, and adjusted to maintain an end-tidal carbon dioxide tension of 35–40 mmHg. The temperature was measured with an esophageal stethoscope and maintained at temperature of at least 35 °C.

During surgery, a 16 or 18 Fr urethral catheter was inserted and the balloon of the urethral catheter was inflated with 10 mL of sterile water. The control group received 20 mL of normal saline and the nefopam group received 20 mg of nefopam mixed with 20 mL of normal saline over a period of 20 min, at 1 h before the end of the operation. After surgery, neostigmine (0.05 mg/kg) and glycopyrrolate (0.01 mg/kg) were used to reverse muscle relaxation, and the patient emerged from anesthesia. Intravenous patient-controlled analgesia with fentanyl alone was connected at the end of anesthesia. Fentanyl (50 μg) as a rescue painkiller was administered at the patient’s request when the postoperative pain score was >3.

The primary outcomes were the severity of CRBD at 20 min, 1 h, 2 h, and 6 h postoperatively using an 11-point numeric rating scale ranging from 0 (no discomfort) to 10 points (most extreme discomfort). The secondary outcomes were the severity of postoperative pain using an 11-point numeric rating scale ranging from 0 (no pain) to 10 points (most extreme pain), the incidence of rescue painkiller usage, the incidence of postoperative nausea and vomiting (PONV), and side effects of nefopam (cardiovascular, digestive, and anticholinergic side effects, including sweating, dry mouth, and tachycardia).

Based on previous study of butylscopolamine [1], assuming that the effect of butylscopolamine on CRBD was similar to that of nefopam, 50 patients were needed for each group, with power = 0.8 and α = 0.05. We decided to include 55 patients per group to allow for a 10% dropout rate.

Data are expressed as means ± standard deviation or numbers. Categorical data were compared using the chi-squared test. Comparison was performed using the independent *t*-test for continuous variables. CRBD and postoperative pain were analyzed using a repeated measures analysis of variance. All statistical analyses were performed using SPSS version 20.0 (IBM Corp., Armonk, NY, USA). In all analyses, *p* < 0.05 was taken to indicate statistical significance.

## 3. Results

A total of 110 male patients were enrolled and allocated randomly into two groups. One patient in the nefopam group was excluded because the surgery was converted to open nephrectomy. Therefore, 109 were analyzed (Figure 1). Demographic data for these 109 male patients are shown in Table 1, and were comparable between the two groups.

The severity score of CRBD in both groups decreased over time. However, the score of CRBD in the nefopam group was significantly lower than that in the control group throughout the study period (Table 2). A comparison of CRBD severity over time using a repeated measures analysis of variance showed significant differences in overall CRBD severity between the two groups (F = 3.527, *p* = 0.017). The postoperative pain score of the control group was significantly higher than that of the nefopam group at 20 min, 1 h, 2 h, and 6 h postoperatively (all, *p* < 0.001) (Table 2). A comparison of postoperative pain scores over time using a repeated measures analysis of variance showed significant differences in overall postoperative pain scores between the two groups (F = 3.877, *p* = 0.011).

Rescue pain killer use was less in the nefopam group than in the control group at 20 min, 2 h, and 6 h postoperatively (Table 3). There were no significant differences in the incidence of PONV or side effects between the two groups (Table 3).

## 4. Discussion

The results of this study indicate that postoperative CRBD was relieved by the intraoperative administration of nefopam (20 mg) before the end of surgery without any side effects. In addition, nefopam (20 mg) reduced postoperative pain and the usage of rescue opioid analgesics. To the best of our knowledge, this is the first paper that clearly separated the surgical pain of the bladder and catheter-related bladder discomfort and demonstrates the effect of nefopam on CRBD, unlike previous papers that targeted patients undergoing bladder surgery [12,13].

In previous studies, several different types of agents were used to reduce CRBD. First of all, antimuscarinic agents, such as oxybutynin [11], tolterodine [7], and gabapentin [4], were reported to lessen the incidence and severity of CRBD, but increase the incidence of antimuscarinic complications, such as dry mouth. Secondly, glycopyrrolate [12] and butylscopolamine [1] were shown to decrease CRBD but can cause tachycardia. Thirdly, tramadol [5], ketamine [14], and dexmedetomidine [2] were significantly effective for the prevention of CRBD. Although there were beneficial effects, these agents have risks, such as sedation, PONV, and postoperative reductions in oxygen saturation.

Nefopam is a non-opioid, centrally acting analgesic derived from the non-sedative benzoxazocine [8]. It is known to act in the brain and spinal cord to relieve pain via an unconventional mechanism. Its main mechanisms of analgesic action are increases in the activity of serotonin, norepinephrine, and dopamine [14,15], and the inhibition of glutamate secretion via the modulation of calcium and sodium channels [16,17,18,19]. In addition, it has anticholinergic or sympathomimetic activity [7,8]. Therefore, nefopam is commonly used as an effective analgesic adjuvant for pain in the perioperative period, and is known to reduce postoperative opioid consumption. The usual intravenous dose of nefopam is 20 mg, and according to the company (Pharmbio Korea Inc, Seoul, Korea) recommendation, repeated administration is possible every 4–6 h if necessary.

CRBD has been shown to be due to involuntary contractions of the bladder caused by catheter-induced bladder irritation, which is mediated by muscarinic receptors [20]. There are many M2 receptors and a few M3 receptors in the bladder wall. M2 receptor activation triggers contraction of the detrusor muscle, and selective M3 receptor inactivation causes M2-mediated detrusor muscle contraction. Therefore, it seems that the antimuscarinic effect of nefopam directly reduces CRBD, consistent with previous studies, in which antimuscarinic agents showed effects on CRBD. Although demonstrated only in animal models, the raphe nuclei in the spinal cord are serotonin-containing terminals. The lumbosacral autonomic nuclei, which act as bladder sphincter motor nuclei, receive serotonin stimuli from the raphe nuclei [21,22]. Therefore, selective serotonin uptake inhibitors are known to affect the raphe nuclei and prevent an overactive bladder [23]. Based on these animal studies, nefopam acting on the spinal cord and the central nervous system may affect serotonin stimuli from the raphe nuclei and reduce CRBD. Although also only demonstrated in animal experiments, dopamine D1 receptor activation suppresses the micturition reflex, while D2 receptor activation has the opposite effect [24]. Therefore, the dopamine activation effect of nefopam may be involved in CRBD relief.

Unlike most previous studies, which were performed in patients undergoing bladder surgery, our patients were enrolled for robotic nephrectomy. Bladder surgery, such as transurethral resection of a bladder tumor or ureter stent insertion after stone removal, can cause mild to moderate suprapubic pain and postoperative pain. We felt that patients may confuse CRBD with these types of pain or CRBD may feel more severe. Therefore, we chose patients undergoing robotic nephrectomy because this procedure has been reported to show lower levels of postoperative pain than conventional surgery [25], and the surgical incision site, the bladder, and the catheter were far apart, so CRBD could be measured more accurately.

Although we used a lower dose than in previous studies (below the minimum clinical dose) [13], nefopam markedly reduced CRBD. Furthermore, in this study, only a single treatment with the minimal dose of nefopam was effective for 6 h after surgery. This was thought to be due to the longer than expected duration of activity of the drug. In a previous study, nefopam was shown to have an effect on pain relief for up to 3 months [26]. In addition, there were no complications because the dose of the drug administered was small.

There are some limitations to our study. There have been no previous animal or human studies regarding the mechanism of how nefopam reduces CRBD, so we could not explain the mechanism precisely in this study. Therefore, further research is needed to determine the mechanism of action of nefopam. Secondly, nefopam was used at the minimal effective dose of 20 mg, so we could not determine the risk of complications at higher doses. Thirdly, it is not possible to exclude the possibility of masking of CRBD due to the usage of opioid analgesics during the study period. Finally, there was a statistically significant difference in CRBD and postoperative pain in this study, but there was no extreme difference in scores. Therefore, it is expected that there will be situations in which statistical differences do not necessarily result in actual clinical differences.

## 5. Conclusions

In conclusion, this study shows that intraoperative administration of nefopam reduced the severity of CRBD and postoperative pain in patients undergoing robotic nephrectomy. Additional investigations are required to gain a better understanding of the use of nefopam for the resolution of CRBD.

## Figures and Tables

**Figure 1 jcm-08-00519-f001:**
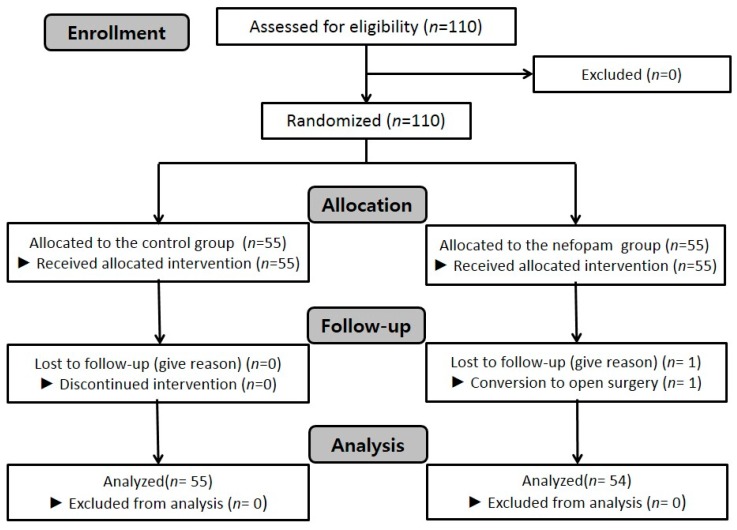
Flow chart of patient enrollment.

**Table 1 jcm-08-00519-t001:** Demographic data.

	Control Group (*n* = 55)	Nefopam Group (*n* = 54)
Age (year)	51.8 ± 11.7	52.6 ± 12.0
Height (cm)	172.0 ± 7.0	171.8 ± 6.0
Weight (kg)	76.5 ± 11.1	74.7 ± 11.5
Body mass index	25.9 ± 3.0	25.3 ± 3.4
Operation time (min)	141.7 ± 42.0	143.9 ± 46.2

**Table 2 jcm-08-00519-t002:** The severity of CRBD and postoperative pain.

	Control Group (*n* = 55)	Nefopam Group (*n* = 54)	95% CI of the Difference	*p*-Value
The severity of CRBD after operation				
20 min	4.8 ± 1.3	4.1 ± 0.8	4.3 to 4.7	0.001
1 h	3.5 ± 1.2	2.3 ± 1.0	2.7 to 3.2	<0.001
2 h	2.7 ± 0.8	1.6 ± 0.8	2.0 to 2.4	<0.001
6 h	2.5 ± 1.0	1.3 ± 0.6	1.7 to 2.1	<0.001
Postoperative pain				
20 min	6.8 ± 0.8	6.2 ± 0.9	6.4 to 6.7	0.001
1 h	5.3 ± 1.4	4.1 ± 1.5	4.4 to 5.0	<0.001
2 h	4.7 ± 1.1	3.5 ± 1.0	3.9 to 4.4	<0.001
6 h	4.4 ± 1.2	3.3 ± 1.3	3.6 to 4.1	<0.001

CRBD: catheter-related bladder discomfort.

**Table 3 jcm-08-00519-t003:** The use of rescue analgesics and incidence of adverse events.

	Control Group (*n* = 55)	Nefopam Group (*n* = 54)	*p*-Value
The use of rescue analgesics after operation			
20 min	55 (100%)	49 (90.7%)	0.027
1 h	29 (52.7%)	30 (55.6%)	0.459
2 h	33 (60.0%)	20 (37.0%)	0.013
6 h	29 (52.7%)	4 (7.4%)	0.000
PONV	2 (3.6%)	2 (3.7%)	0.985
Side effects	0 (0.0%)	0 (0.0%)	1.000

PONV: postoperative nausea and vomiting.

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
