# Peer review of "Effects of Intraoperative Nefopam on Catheter-Related Bladder Discomfort in Patients Undergoing Robotic Nephrectomy: A Randomized Double-Blind Study"

_jcm, 2019, doi:10.3390/jcm8040519_

Round 1

Reviewer 1 Report

The authors should present how many patients were male and how many female and whether there was a difference in findings. As they stated themselves in the Introduction, there is a discrepancy in catheter-related bladder pain among both sexes.

In general, this is a very nice study, most impressive is the decreased use of rescue analgesics at 2 and 6 hours in nefopam group.

Author Response

[12, APRIL, 2019]

 Dear The Journal of Clinical Medicine Editorial Board

First of all, I, along with my coauthors, thank the editors and reviewers who reviewed the paper.

 As you mentioned, there is a discrepancy in catheter-related bladder pain among both sexes, so this study only targeted male patients. To add more clarity to our readers on this point, we corrected the phrase that we were targeting only male patients. (line 19, 50)

 Next, the average values and P-values specified on the 88-92 line of the Method section have been deleted and the sentence has been modified. (line 89)

 Third, there were statistical differences but there was no extreme difference between the two groups in numbers. Therefore, we inserted this mension in the limitation session of the discussion that it can not always be linked to actual clinical difference. (line 181-184)

 Major spelling errors have been corrected. (line 38, 39, 41, 42, 51, 93, 130, 132-134, 151, 177, 179, and 180)

 The email addresses of co-authors incorrectly written. has been corrected. (line 14)

 Finally, we have added a blank statement of informed consent as an attachment file.

 Sincerely,

 Eun-su Choi

Department of Anesthesiology and Pain Medicine

Korea University Ansan Hospital, Gyeonggi-do, Republic of Korea

Reviewer 2 Report

This is a well-designed and well-written work presented here by Chi-Bum, et al.

My comments are as follows:

-Data presentation in Methods part and talking about numbers and p value should not typically be there-Line 89-92.

-The important point here which should be mentioned here when it comes to pain interpretation is that the statistical difference may not necessarily mean clinical difference. 

Author Response

[12, APRIL, 2019]

 Dear The Journal of Clinical Medicine Editorial Board

 First of all, I, along with my coauthors, thank the editors and reviewers who reviewed the paper.

 As you mentioned, there is a discrepancy in catheter-related bladder pain among both sexes, so this study only targeted male patients. To add more clarity to our readers on this point, we corrected the phrase that we were targeting only male patients. (line 19, 50)

 Next, the average values and P-values specified on the 88-92 line of the Method section have been deleted and the sentence has been modified. (line 89)

 Third, there were statistical differences but there was no extreme difference between the two groups in numbers. Therefore, we inserted this mension in the limitation session of the discussion that it can not always be linked to actual clinical difference. (line 181-184)

 Major spelling errors have been corrected. (line 38, 39, 41, 42, 51, 93, 130, 132-134, 151, 177, 179, and 180)

 The email addresses of co-authors incorrectly written. has been corrected. (line 14)

 Finally, we have added a blank statement of informed consent as an attachment file.

 Sincerely,

 Eun-su Choi

Department of Anesthesiology and Pain Medicine

Korea University Ansan Hospital, Gyeonggi-do, Republic of Korea

J. Clin. Med. EISSN 2077-0383 Published by MDPI AG, Basel, Switzerland RSS E-Mail Table of Contents Alert
Back to Top